# Thyroid Malignancy and Cutaneous Lichen Amyloidosis: Key Points Amid *RET* Pathogenic Variants in Medullary Thyroid Cancer/Multiple Endocrine Neoplasia Type 2 (MEN2)

**DOI:** 10.3390/ijms25189765

**Published:** 2024-09-10

**Authors:** Laura-Semonia Stanescu, Adina Ghemigian, Mihai-Lucian Ciobica, Claudiu Nistor, Adrian Ciuche, Andreea-Maria Radu, Florica Sandru, Mara Carsote

**Affiliations:** 1PhD Doctoral School, “Carol Davila” University of Medicine and Pharmacy, 0505474 Bucharest, Romania; laura-semonia.stanescu@drd.umfcd.ro; 2Department of Clinical Endocrinology V, C.I. Parhon National Institute of Endocrinology, 011863 Bucharest, Romania; adina.ghemigian@umfcd.ro (A.G.); carsote_m@hotmail.com (M.C.); 3Department of Endocrinology, “Carol Davila” University of Medicine and Pharmacy, 020021 Bucharest, Romania; 4Department of Internal Medicine and Gastroenterology, “Carol Davila” University of Medicine and Pharmacy, 020021 Bucharest, Romania; 5Department of Internal Medicine I and Rheumatology, “Dr. Carol Davila” Central Military University Emergency Hospital, 010825 Bucharest, Romania; 6Department 4—Cardio-Thoracic Pathology, Thoracic Surgery II Discipline, “Carol Davila” University of Medicine and Pharmacy, 0505474 Bucharest, Romania; adrian.ciuche@umfcd.ro; 7Thoracic Surgery Department, “Dr. Carol Davila” Central Emergency University Military Hospital, 010825 Bucharest, Romania; 8Department of Dermatovenerology, Elias University Emergency Hospital, 011461 Bucharest, Romania; andreea.radu@rez.umfcd.ro (A.-M.R.); florica.sandru@umfcd.ro (F.S.); 9Department of Dermatovenerology, “Carol Davila” University of Medicine and Pharmacy, 020021 Bucharest, Romania

**Keywords:** thyroid, RET, amyloidosis, neuroendocrine, multiple endocrine neoplasia, skin, thyroid malignancy, lichen, codon, gene

## Abstract

We aimed to provide an updated narrative review with respect to the *RET* pathogenic variants and their implications at the clinical and molecular level in the diagnosis of medullary thyroid cancer (MTC)/multiple endocrine neoplasia (MEN) type 2, particularly with respect to the presence of cutaneous lichen amyloidosis (CLA). We searched English-language, in extenso original articles with no timeline nor study design restriction that were published on PubMed. A traditional interplay stands for CLA and MTC in MEN2 (not MEN3) confirmation. While the connection has been reported for more than three decades, there is still a large gap in understanding and addressing it. The majority of patients with MEN2A-CLA have *RET* pathogenic variants at codon 634; hence, it suggests an involvement of this specific cysteine residue in both disorders (most data agree that one-third of C634-positive subjects have CLA, but the ranges are between 9% and 50%). Females seem more prone to MEN2-CLA than males. Non-C634 germline *RET* pathogenic variants included (at a low level of statistical evidence) the following: *RET* V804M mutation in exon 14 for MTC-CLA (CLA at upper back); *RET* S891A mutation in exon 15 binding *OSMR* variant G513D (familial MTC and CLA comprising the lower legs to thighs, upper back, shoulders, arms, and forearms); and C611Y (CLA at interscapular region), respectively. Typically, CLA is detected at an early age (from childhood until young adulthood) before the actual MTC identification unless *RET* screening protocols are already applied. The time frame between CLA diagnosis and the identification of *RET* pathogenic variants was between 5 and 60 years according to one study. The same *RET* mutation in one family is not necessarily associated with the same CLA presentation. In MTC/MEN2 subjects, the most affected CLA area was the scapular region of the upper back. Alternatively, another hypothesis highlighted the fact that CLA is secondary to long-term prurit/notalgia paresthetica (NP) in MTC/MEN2. *OSMR* p. G513D may play a role in modifying the evolutionary processes of CLA in subjects co-harboring *RET* mutations (further studies are necessary to sustain this aspect). Awareness in CLA-positive patients is essential, including the decision of *RET* testing in selected cases.

## 1. Introduction

Medullary thyroid carcinoma (MTC), a rare neuroendocrine tumor, involves 1–2% of thyroid cancers (SEER—Surveillance, Epidemiology, and End Results) [1,2,3,4]. This rate is lower than 3–5% as previously shown, due to a relative increase in papillary-type (PTC) incidence over the last three decades [4], noting that differentiated thyroid malignancies remain the most frequent histological types all over the world nowadays [5,6,7].

We aimed to provide an updated narrative review with respect to the *RET* pathogenic variants and their implications at the clinical (dermatology and endocrinology assessments) and molecular level in the diagnosis of MTC/multiple endocrine neoplasia (MEN) type 2, particularly with respect to the presence of cutaneous lichen amyloidosis (CLA). We searched English-language (at least at the abstract level), in extenso original articles with no timeline nor study design restriction that were published on PubMed regarding the key points of the mentioned thyroid malignancy and its skin signature of the amyloid type.

## 2. Medullary Thyroid Carcinoma (MTC) and Multiple Endocrine Neoplasia (MEN) Harboring *RET* Pathogenic Variants

MTC may occur sporadically (sMTC) (75% of all cases) or hereditary as a manifestation of MEN type 2 (formerly named MEN2A) and MEN 3 (formerly called MEN2B) [8,9]. MEN2 has a prevalence of 1 in 25,000 people and accounts for over 95% of the hereditary MTC [10,11]. Four clinical variants of MEN2 have been described with various multidisciplinary elements on first presentation and long-term surveillance [12,13,14]. Firstly, classic MEN2A, featuring MTC in 100% of the patients and less frequent occurrence of phaeochromocytoma (PHEO) in 50% of the subjects, or primary hyperparathyroidism (HPTH) in 15% of the MEN2 individuals, or both; secondarily, MEN2A with CLA; thirdly, MEN2A with Hirschsprung disease (HD); and fourthly, isolated familial MTC (FMTC) that accounts for about ~15% of the patients with hereditary MTC and it is diagnosed only if MTC is present in the family members as well [1,15]. On the other hand, MEN3 involves a rare syndrome, also including MTC (5% of hereditary MTCs), etc. [11,16,17].

MTC is related to the Rearranged during Transfection (*RET*) gene [18,19,20,21,22,23,24,25,26,27] (Figure 1).

RET functions are numerous [28,29,30,31,32,33,34,35,36,37,38,39,40] (Figure 2).

In the absence of the ligand, the RET protein is a single unphosphorylated tyrosine kinase receptor, while in cancer cells, *RET* proto-oncogene led to the auto-phosphorylation of the tyrosine residues [41,42,43].

Some *RET* pathogenic variants in thyroid malignancies are inherited and are present throughout every cell in the body (germline type) and others are acquired (somatic type) that are present only in certain cells in the body [44,45]. Even so, mutational screening is mandatory in all patients with MTC, allowing the detection of germline mutations in initially “so-called” sporadic MTC in up to 6.5% of the patients [46,47]. The etiology of sMTC has not been completely elucidated at this point [48,49,50,51,52,53,54,55,56] (Figure 3).

HD occurs in approximately 7% of MEN2 subjects; conversely, 2% to 5% of the patients diagnosed with HD have MEN2 [57,58,59,60]. *RET* pathogenic variants in individuals identified with MEN2 and HD are point mutations involving codons in exon 10: 609 (15%), 611 (5%), 618 (30%), and 620 (50%) [41,53,55]. It seems paradoxical that MEN2 and HD may occur together, since *RET* mutations associated with HD are loss-of-function, while those associated with MEN2A are gain-of-function. This dual occurrence could be explained by the fact that constitutive activation of *RET* is sufficient to trigger the neoplastic transformation of the C cells (at the level of thyroid cancer), yet insufficient to generate a trophic response in the precursor neurons due to a lack of expression of the RET protein at the cell surface [61,62,63]. Of note, the congenital absence of the ganglion cells at the level of the myenteric and submucosal intestinal plexus (namely, HD) has been found amidst other syndromes such as trisomy 21, congenital central hypoventilation syndrome, and Mowat–Wilson syndrome, while (other than *RET* proto-oncogene) endothelin receptor type B (*EDNRH*) gene was the most frequent gene incriminated in HD [63].

In isolated FMTC, the most common variants affect codons 768, 790, or 804 (exons 13 or 14), with 804 being the most frequent [11,15,64,65]. MEN3 is exclusively related to exon 16 (M918T). These mutations lead to increases in ATP-binding and auto-phosphorylation activity, thereby mediating a dimerization-independent activation of RET kinase [11,15,66,67,68]. In less than 10% of the patients, MEN3 is associated with an A883F mutation or with double mutations, which include V804M and either Y806C, S904C, E805K, or Q781R [69,70,71,72].

Current guidelines recommend germline *RET* testing in all patients with a new diagnosis of MTC, including family members. As a result of the link between the type of *RET* pathogenic variant and the aggressiveness of CMT in children with mutations at codon 918, exon 16 (harboring the highest risk) should undergo thyroidectomy in their first year of life. Those with mutations at codon 634, exon 11 or codon 883, or exon 15 (that are regarded as high risk) should have a thyroidectomy performed at the age of 5 years or earlier if the serum levels of calcitonin become elevated. In children with any of the other *RET* mutations (moderate risk), the decision regarding the age of prophylactic thyroidectomy is no longer based upon genotype alone, but is currently driven by the hormonal panel, specifically, by the increasing trend of the serum calcitonin levels [11,15].

## 3. Insights into Lichen Amyloidosis (LA)

Amyloidosis, characterized by abnormal misfolded protein-based deposits in addition to amyloid fibrils, affects various organ functions. It is divided into cutaneous and systemic forms (that may be accompanied by different co-morbidities) based on localization, but it can be sub-grouped via its biochemical structure [73,74,75]. The cutaneous form of amyloidosis was first recognized in the 1930s by Freudenthal [76,77], who noted the presence of Congo red-positive hyaline bodies within the epidermis [76,77]. The deposition of amyloid in previously apparently normal skin without deposits in the internal organs is known as primary localized cutaneous amyloidosis (PLCA), and secondary amyloidosis, which forms on a pre-existing dermatologic lesion (benign or malignant) [73,75]. Secondary PLCA is caused by rubbing/scrubbing (“friction amyloidosis”) [76,77,78]. Its subtypes are macular (MA), lichen (LA)—the most common, and nodular (NA); biphasic amyloidosis (BA) is MA + LA [79,80,81,82,83,84]. The most common site involves the extensor surfaces of the extremities [85]. Clinically, it is difficult to distinguish different subtypes of PLCA, with the histological and immunohistochemistry characteristics being a very important tool for diagnosis. MA, LA, and BA are keratinocyte-derived, in which cytokeratins serve as amyloid precursors, while NA involves an immunoglobulin that is light-chain-derived and it is associated with dermal plasma cell infiltration [86,87,88]. With regard to LA, the amyloid deposits display immunoreactivity to antikeratin antibodies, but not to antibodies against A protein, pre-albumin, or fibronectin [78,88].

Generally, amyloid stands on >20 proteins [89,90,91]. In nature, proteins usually have both an alpha helix and β sheet structure. In amyloid, electron microscopy studies have shown the abnormal folding of proteins that are arranged predominantly as β-pleated sheets over the alpha helix [92,93,94]. Amyloidosis manifests when these sheets are misfolded, resulting in extracellular deposits. As for why these deposits form, the exact mechanism is not fully understood nowadays. Potential causes are as follows: protein over-production makes them fold into β-pleated sheets, abnormal protein structure, other contributors like low pH, metal ions, etc. [95,96,97]. Normally, certain mechanisms degrade the abnormal proteins but if they fail, amyloid can deposit in tissue. This process is highly variable and a lag phase is described in which the prerequisites are present, but no amyloid fibrils are formed yet. This lag phase can last from weeks to years. After a nucleus of amyloid is formed, aggregation occurs under fast kinetics and the development of the insoluble architecture soon follows [98,99,100].

Overall, the pathogenesis is considered multifactorial, involving both environmental and genetic factors. Regarding environmental factors, it is worth mentioning that Epstein–Barr virus (EBV) has been demonstrated in a high percentage of cases of MA, LA, or BA. The presence of EBV in the epidermis has been postulated as a contributing event in the degeneration of keratinocytes [101,102,103]. Recently, COVID-19 infection was suspected to exacerbate amyloidosis, potentially acting as a supplementary contributor or as a trigger for pre-existing histological lesions (as seen in other associated diseases following a coronavirus infection) [104,105,106,107]. When assessing if certain factors predispose individuals to develop PLCA, familial forms have been described; PLCA is common in Southeast Asia, South America, and Middle Eastern regions [108]. In addition, up to 10% of all PLCA cases are familial and an autosomal dominant inheritance may be observed [109,110]. Racial susceptibility and familial aggregation suggest that underlying genetic factors must play a role in the pathogenesis of PLCA. In cases of familial PLCA, pathogenic heterozygous missense mutations have been mapped to the oncostatin M receptor beta subunit (*OSMR*) gene [111,112,113,114,115,116,117,118,119,120,121] (Figure 4).

## 4. CLA in MTC/MEN2

### 4.1. Traditional Pathogenic Aspects in CLA in Patients Diagnosed with MEN2

Traditionally, scratching appears to play an important role in the pathogenesis of CLA by causing epidermal cell damage that leads to filamentous degeneration of the keratinocytes, subsequent apoptosis, and conversion of the filamentous masses into typical amyloid material. This mechanism is supported by the fact that LA commonly presents as a pruriginous lesion. [121,122]. With regard to the majority of LA cases that are associated with MEN2, pruritus precedes the skin lesion, causing subsequent scratching and therefore cutaneous lesions. In an early study conducted by Chabre et al. [122] in 1992, a French MEN2A family (as the term was used at that moment) had three members with the same type of pruritic scapular skin lesion associated with paresthesia and hyperalgesia in the same area. Despite extensive analysis of the skin biopsies (special staining and immunohistochemistry analysis), no amyloid deposits were found at that time [122]. Moreover, these previous findings suggested a neurological origin of the cutaneous manifestation associated with MEN2A. Based on this hypothesis, CLA in MEN2 may be caused by a neurological condition called notalgia paresthetica (NP) [122,123,124,125,126]. NP refers to a neuropathy involving the dorsal primary divisions of spinal nerves, which causes localized pruritus. It was first described in 1934 and afterwards reviewed by Weber and Poulos [127,128], who extended the term to several entities previously described as “localized pruritus”, “puzzling posterior pigmented patches”, and “peculiar spotty pigmentation” [123,124,125,126]. The main symptom of NP is pruritus, and it is typically unilateral, localized on two-thirds of the scapula. In patients with long-standing pruritus, secondary lesions such as hyperpigmentation and lichen development may be visualized. The relationship between NP and MEN2 might be explained by the fact that neural crest cells, implicated in the embryological development of the adrenal medulla and para-follicular C cells of the thyroid, are also involved in the embryogenesis of the thoracic sensory fibers that explains the pruritus sites [127,128].

Another hypothesis relates to the *RET* ligand, namely, GDNF family receptor alpha (GFRα) proteins with a potential role in enteric neurons, parasympathetic and sympathetic neurons, and sensory neurons, but further evidence is necessary [30,129,130,131,132,133,134,135].

### 4.2. Sample-Based Data in MTC/MEN2-Related CLA

After Gagel et al. [136] described for the first time CLA in MEN2 (in 1989), it was found that multiple papules disposed as a well-demarcated plaque in the scapular area might accompany the endocrine malignancy as a skin signature (other than the presence of a carcinoid syndrome in metastatic MTC) [136]. Although the association of CLA with MEN2 has long been established, the reason why only a small percentage of MEN2 patients actually develop this manifestation remains an open matter nowadays. MEN2-related CLA occurs in patients with C634 *RET* pathogenic variants, and it has been estimated that more than 30% of the patients with this type of mutation will be CLA-positive across their life span [56,137]. Verga et al. [120] reported that CLA was found only in MEN2A/FMTC families harboring the *RET* pathogenic variant in codon 634, with an incidence of 36% (9/25 of the affected subjects), mentioning that two patients actually lacked these skin lesions, but the symptom of the neurological pruritus (namely, NP) in the upper back was present [120]. Moreover, in a cohort of 38 MEN2A cases harboring the C634 mutation, Scapineli et al. [121] reported CLA in 50% of these individuals, with the C634R mutation being the most prevalent (7 kindred, 35%) [121]. A smaller percentage was reported by Eng et al. [52], specifically, CLA was detected in approximately 9% (18/199) of MEN2 individuals, and all 18 families carried the *RET* codon 634 pathogenic variant [52]. Qi et al. [138] showed that the leading mutation at codon 634 in exon 11 was C634Y (affecting 43.6%, 24/55 of the subjects), followed by C634R (27.3%, 15/55 individuals), C634W (10.9%, 6/55), C634G (9.1%, 5/55), C634F (1.8%, 1/55), respectively, and C634 (5.5%, 3/55) [138]. The fact that the majority of the subjects diagnosed with MEN2A-related CLA harbor codon 634 suggests an involvement of this specific cysteine residue in both endocrine and dermatologic disorders and this might be a key turning point in understanding common pathogenic features [52,138].

Yet, other *RET* germline mutations of the intracellular TKD have been described in MEN2-related CLA; for instance, Rothberg et al. reported a case of an American female with the germline *RET* V804M mutation within exon 14 with MTC and CLA on the upper back [56]. In a Chinese family with FMTC and CLA comprising the lower legs to thighs, upper back, shoulders, arms, and forearms, Qi et al. described (in 2015) the *RET* S891A mutation within exon 15 binding *OSMR* variant G513D [139]. In 2018, Qi et al. described a novel genotype–phenotype relationship between MEN2A and CLA. They reported a Chinese pedigree with 17 individuals carrying the C611Y *RET* mutation with one member (1/17, representing 5.9%) affected by CLA in the interscapular region [140].

To date, no family with MEN3 has been reported to be diagnosed with CLA. Moreover, families with CLA (but without MEN disease features) do not seem to harbor *RET* pathogenic variants [11,121,141]. To summarize, the genetic basis for MEN2A-related CLA remains obscure, with it being suggested that the primary cause is actually neurogenic, and the skin lesions are only a secondary phenomenon due to chronic pruritus and repeated scratching (as we already mentioned) [11,121,122,141].

Other potential contributors to MEN2-CLA have been studied. Gender-related predominance in the prevalence of CLA was observed by some authors. For instance, Qi et al. [139,140] described a higher prevalence in the female population with a male-to-female ratio of approximately 2:9 (6:27), respectively, of 1.0:3.6 (12:43) [139,140]. Similarly, Scapianeli et al. [121] described a CLA prevalence of 1.0/2.3 in men/women in the mentioned cohort [121].

Interestingly, many literature data describe the presence of CLA earlier than other endocrine (clinical) elements of MEN2, thus delaying the diagnosis of MEN2-related CLA. For example, Qi et al. revealed that the mean age at diagnosis of CLA with the *RET* mutation was 29.5 years (range between 5 and 60 years) [140]. The age at onset for CLA for most MEN2 individuals was often in infancy or adolescence and most patients presented pruritic symptoms before the actual endocrine diagnosis of MTC [11,120,140,142,143,144,145,146]. Individuals with the same or different *RET* pathogenic variant typically presented with a variable clinical manifestation of CLA. The most described phenotypes are in the scapular region of the upper back, exhibiting on the side, midline, or bilateral extending across the midline, followed by the hyperpigmentation, respectively, by the papules developed in the same area after many years of itching and scratching [11,120,140,143,144,145,146]. Thus, much attention should be paid to the recognition of CLA located in the scapular region. The most recent guidelines for MTC recommend that subjects with CLA located in the scapular region should be investigated clinically and then undertake germline *RET* screening for MEN2 [11,15]. A more expanded affected region than previously described, encompassing the upper back, shoulders, arms, and legs manifesting as BA, harbored the *RET* S891A pathogenic variant and *OSMR* variant G513D in a Chinese FMTC family, as mentioned [139]. Conversely, MEN2-related CLA was strictly located in the scapular region and had the *RET* C634F/G mutation without the *OSMR* mutation in four Chinese patients [140]. Therefore, *OSMR* p.G513D may play a role in modifying the evolutionary process of CLA with *RET* pathogenic variants [139,140] (Table 1).

## 5. Discussion

According to the searched data [56,121,128,136,139,142,143,144,145,146,147,148,149,150,151,152,153,154,155,156], a traditional interplay stands for CLA and MTC in MEN2 (not MEN3) confirmation. While the connection has been reported for more than three decades, there is still a large gap in understanding and addressing it from a practical multidisciplinary point of view and awareness is essential, especially since the timeline perspective includes an early CLA identification before the recognition of the thyroid cancer in subjects who are not already confirmed with *RET* pathogenic variants or they are not under endocrine surveillance protocols. Females seem more prone to MEN2-CLA than MEN2 males. The majority (but not all) of patients with MEN2A-related CLA have *RET* pathogenic variants at codon 634; hence, this represents a key point that suggests an involvement of the specific cysteine residue in both disorders, which may imply the possibility of additional mechanisms in the pathogenic signal transduction pathways for MEN2A-related CLA. Additionally, *OSMR* p.G513D may play a role in modifying the evolutionary process of CA in subjects co-harboring *RET* mutations (but the level of statistical evidence remains low in this specific matter) [56,121,128,136,139,142,143,144,145,146,147,148,149,150,151,152,153,154,155,156]. 

### 5.1. Unraveling Clinical CLA Insights

Regardless of the synchronous/asynchronous identification of MEN2 or confirmation of *RET* pathogenic variants at screening genetic testing on one patient, the diagnosis of PLCA should be confirmed by the histological exam in daily practice, whereas the deposition of eosinophil, amorphous globular material in the papillary dermis, melanophages, and increased pigmentation of the basal layer are typically observed [114]. The special stains routinely employed in the detection of amyloid deposits are Congo red and Crystal violet. Also, other stains such as Thiofavin T, periodic acid–Schiff method, and Sirius red may be used for a more defined visualization [80,85,109,114,157]. Of note, Congo red plays a particular role in the pathological confirmation of MTC as well [158,159,160]. Both macular and lichen amyloidosis display amyloid deposits restricted to the upper dermis, particularly the papillary dermis [80,85,109]. As previously reported by Nunziata et al., staining for amyloid may be negative in a skin biopsy and multiple biopsies are needed to demonstrate amyloid content [156]. Such negative results may be explained by the fact that in the initial stages of PLCA, there is a small amount of amyloid deposit. Perhaps, dermal amyloid might be demonstrated only after long-term scratching when the skin appears clearly hyperkeratotic and pigmented [80]. Currently, most of these techniques have lost some use in daily practice due to the availability of specific types of antibodies via immunohistochemistry analysis [161]. Electronic microscopy plays a limited role in the everyday diagnosis of CLA. Nevertheless, it might prove a useful tool when the mentioned special stains fail to demonstrate the amyloid deposit, as explained by [161,162].

To date, several treatment options for PLCA have been described: both topical and systemic agents as well as phototherapy and laser therapy. However, with few clinical trials available so far, there is no gold standard treatment. Despite the multitude of treatment options, none are curative and most of them aim to break the itch–scratch cycle. One treatment option could be systemic retinoid treatment, which revealed benefits in decreasing pruritus and the size of the cutaneous lesion in LA and BA [117,163]. Other systemic options were cyclophosphamide and colchicine, which showed benefits on pruritus, pigmentation, and decrease in the size of the lesions with few side effects according to some authors [164]. In cases of refractory PLCA, methotrexate was proposed as an alternative therapy in resolving skin manifestations and also in reducing pruritus [164,165]. Amitriptyline, a tricyclic antidepressant, was indicated in neuropathic itching and it has proved beneficial in the resolution of pruritus in PLCA with no change in the skin lesions [164]. Mild cases of PLCA responded to a topical treatment such as the application of capsaicin to some extent [165]. Topical corticosteroids have demonstrated long-term resolution in LA lesions and improvement in pruritus when combined with topical salicylic acid 25% ointment [166,167]. Dimethyl sulfoxide has been used in LA with varying degrees of success and notable side effects [117]. Alternatively, topical vitamin D3 analogue calcipotriol, the calcineurin inhibitor cyclosporine, menthol, and tacrolimus have been used [117]. Recently, laser therapy was found to be efficient in PLCA (carbon dioxide laser and erbium:yttrium aluminum garnet laser) [168,169]. Different phototherapeutic modalities have also been described in the treatment of PLCA such as UVB irradiation and topical psoralen with UVA (PUVA) therapy with heterogeneous results until the present time [170,171]. Furthermore, additional interventions such as transcutaneous nerve stimulation and surgical procedures (such as electrodessication, dermabrasion) in LA have also been reported [117]. Currently, we have no statistically significant data to pinpoint whether MEN2-CLA should be approached differently than non-MTC cases and whether LCA in association with a thyroid malignancy/neuroendocrine neoplasm is more severe apart from an increased disease burden that comes with the clinical expression due to the *RET* pathogenic variants.

### 5.2. Integrating MTC/MEN2-Related CLA into the Larger Frame of Skin Lesions in Neuroendocrine Neoplasia and Endocrine Tumors/Malignancies

Rarely in the general population are there patients presenting with skin lesions that reflect an underlying endocrine disorder, typically a functionally active tumor. As seen in MEN2-CLA, awareness remains the key operating perspective. For instance, necrolytic migratory erythema (NME) was firstly described by Becker et al. (Becker’s nevus) in 1942 in a woman with an islet cell type of pancreatic carcinoma that was revealed only post-mortem [172]. Further on, a connection with glucagonoma was established, regardless of if the neoplasia was a part of MEN1 in some patients (in addition to other endocrine tumors in the pituitary, parathyroid, thyroid, and adrenal glands) [173,174]. McCune–Albright syndrome (somatic activating mutations in *GNAS* gene) has cafe-au-lait spots, hyper-function of multiple endocrine glands, etc.; autonomous hyper-function most commonly involves the ovary, but also the thyroid (nodular hyperplasia with thyrotoxicosis), adrenal (multiple hyperplastic nodules with Cushing’s syndrome), and pituitary glands (somatotropinomas and prolactinomas), and parathyroid tumors causing HPTH (another type of hereditary HPTH as found in MEN2) [175,176]. As mentioned, MEN1, with an autosomal dominant predisposition to tumors of the parathyroid glands, anterior pituitary, and pancreatic islet cells, associates with multiple cutaneous lesions such as angiofibromas, lipomas, and collagenomas, and, potentially, an increased risk of other non-endocrine malignancies [177,178]. The complex of spotty skin pigmentation, myxomas, endocrine gland over-activity due to various tumors, and schwannomas, namely, Carney complex, stands for an autosomal dominant syndrome involving the skin in terms of developing myxomas (that may have different non-cutaneous sites), and spotty pigmentation [179,180].

Finally, one of the most important aspects in the complex panel of skin signature with respect to the endocrine malignancies/neuroendocrine neoplasia is represented by the carcinoid syndrome in metastatic MTC. This is caused by the liver metastases that interfere with the serotonin (5-hydroxytriptamine), respectively, 5-hydroxyindolacetic acid metabolism as similarly seen in cancers of other primary origins [181,182]. Cutaneous involvement such as episodic flushing is the clinical hallmark in 85% of the patients. It primarily involves the face, neck, and upper chest, which become red to violaceous or purple, and are associated with a mild burning sensation [181,183,184]. Thus, CLA in MTC/MEN2 subjects should be regarded via the larger and complex frame of endocrine and skin interplay that involves a common genetic, epigenetic, molecular, immune, and hormonal background depending on the specific circumstance.

### 5.3. Current Limits and Further Expansion

We are aware of the limits of a non-systematic review, but, noting the current data available, we chose a more flexible approach under various perspectives, from genetic interplay to the clinical expression. There is still a matter of debate: if CLA is more severe in MEN2 versus non-MEN2 subjects; if MTC displays a more severe outcome in CLA-positive versus CLA-negative patients; if there is a distinct risk of non-MTC MEN2 components other than the indications provided by the mutation map in CLA-positive individuals. Overall, early diagnosis of CLA should raise the question of a thyroid malignancy across the life span. Moreover, prurit before the actual skin lesion might pinpoint a pathogenic contributor to lichen, other than the actual *RET* gene influence. We still need large, longitudinal studies to address this crossroad between CLA and MEN2. Although the association of CLA with MEN2 has long been established, the reason why a small percentage of MEN2 patients actually develop this manifestation remains an open matter.

## 6. Conclusions

CLA in thyroid cancer, particularly, MTC (and mostly across the confirmation of MEN2 underlying the *RET* pathogenic variants) stands for the following key points:MEN2-related CLA occurs in patients with C634 *RET* pathogenic variants; most data agree that one-third of C634-positive subjects have CLA, but the ranges are between 9% and 50%.One single study showed the *RET* codons map (codon 634 in exon 11) in CLA: C634Y (affecting 43.6% of the subjects), followed by C634R (27.3%), C634W (10.9%), C634G (9.1%), C634F (1.8%), and C634 (5.5%).Non-C634 germline *RET* pathogenic variants included (at a low level of statistical evidence) the following: *RET* V804M mutation in exon 14 for MTC and CLA (CLA at upper back); *RET* S891A mutation in exon 15 binding *OSMR* variant G513D (FMTC and CLA comprising the lower legs to thighs, upper back, shoulders, arms, and forearms); and C611Y (CLA at interscapular region).Awareness in CLA-positive patients is essential, including the decision of *RET* testing in selected cases (Table 2).

## Figures and Tables

**Figure 1 ijms-25-09765-f001:**
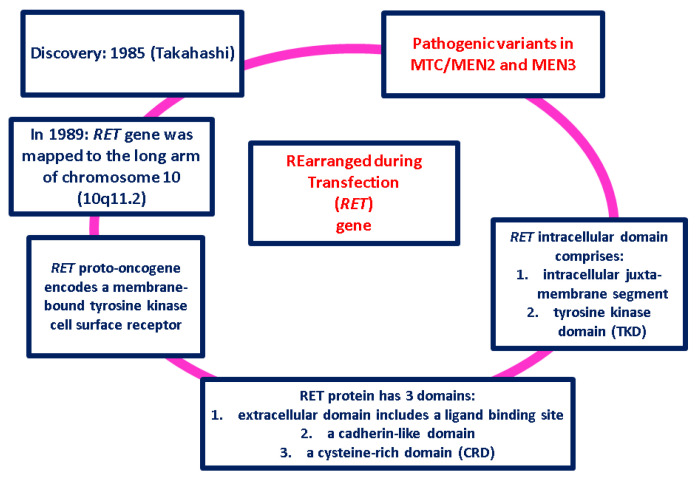
Main highlights with regard to *RET* gene [18,19,20,21,22,23,24,25,26,27].

**Figure 2 ijms-25-09765-f002:**
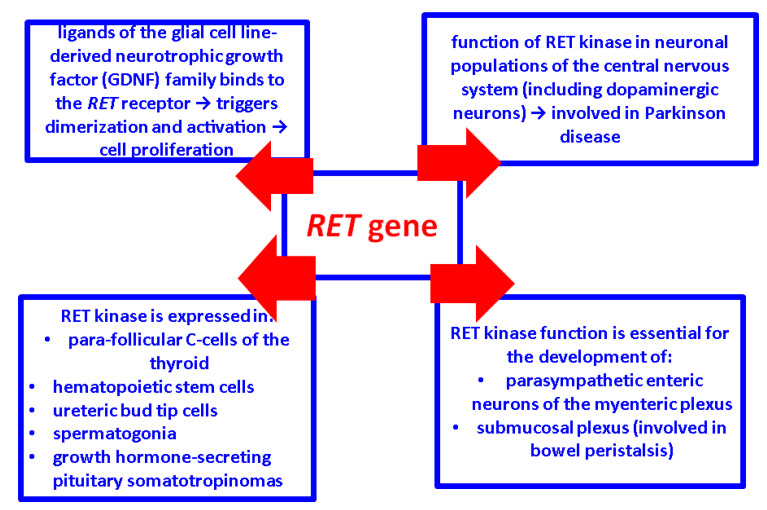
RET kinase functions [28,29,30,31,32,33,34,35,36,37,38,39,40].

**Figure 3 ijms-25-09765-f003:**
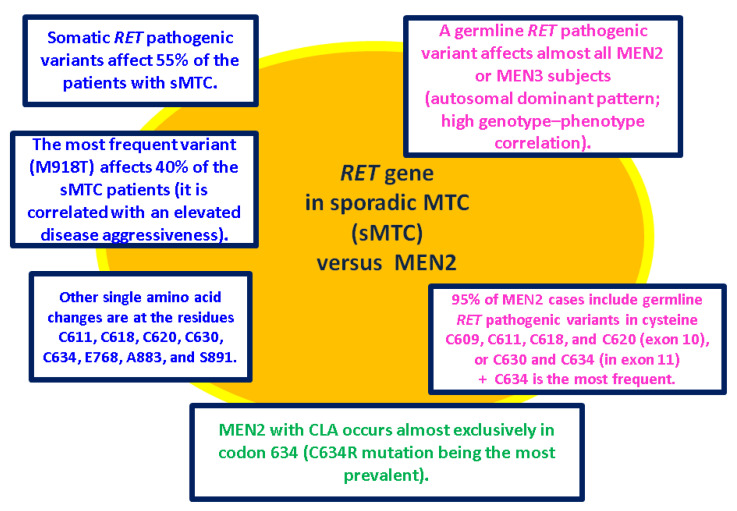
*RET* in sMTC versus MEN [48,49,50,51,52,53,54,55,56].

**Figure 4 ijms-25-09765-f004:**
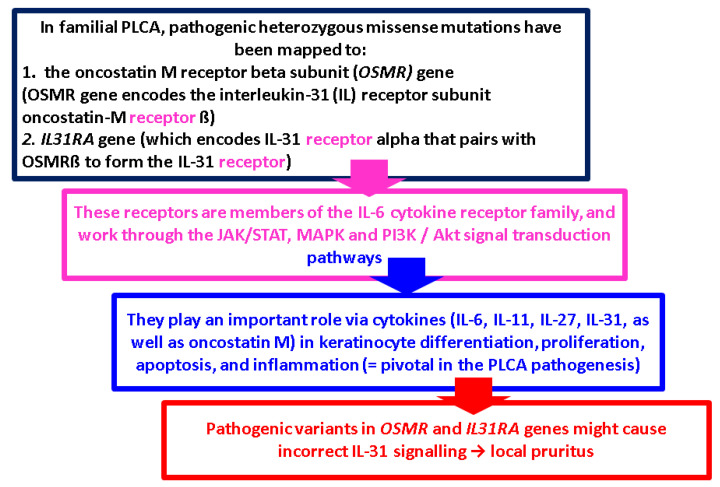
*OSMR* and *IL3RA* genes and pruritus [111,112,113,114,115,116,117,118,119,120,121].

**Table 1 ijms-25-09765-t001:** Sample-focused analysis on published data with respect to MTC/MEN2 and CLA according to our methods (PubMed search based on key terms “lichen amyloidosis” and “thyroid cancer” without any timeline restriction); the display starts with the most recent publication date [56,121,128,136,139,142,143,144,145,146,147,148,149,150,151,152,153,154,155,156]. (Abbreviations: CLA = cutaneous lichen amyloidosis; F = female; GC = glucocorticoids; HPTH = primary hyperparathyroidism; PHEO = pheochromocytoma; MEN = multiple endocrine neoplasia; MTC = medullary thyroid carcinoma; M = male; NA = not available; n = number of family members/affected patients with MEN2A; a = two patients were not evaluated for the presence of CLA; b = all three patients with CLA had the *RET* p.S891A mutation and a novel *OSMR* variant p.G513D, which provides a possible new insight into the mechanism underlying FMTC/CA; c = data expressed as mean and standard deviation for age; of note, the data are displayed across a heterogeneous spectrum depending on the original reports).

Publication Data Reference	Studied Population (MEN Subtype)	RET Mutation	CLA + [n (%)]	Sex CLA + M:F (F%)	Age CLA at Diagnosis (Years)	Clinical Features of CLA	Treatment	Associated Endocrine Lesions MTC [n (%)] PHEO [n (%)] HPTH [n (%)]	Other Observations
Fang X et al., 2022 [147]	28 MEN2A	C634G/F/R/S/W and C611Y	8/28	2:6	18.4 ± 4.6	Interscapular region	NA	NA	Incidence of CLA in C611Y lower than those in C634G/F/R/S/W
Tang HX et al., 2021 [148]	8 MEN2A	C634R	1/8	0:1 (100)	NA	Interscapular area of the left back corresponding to dermatomes T2-T6 level	Topical GC	PHEO, MTC, HPTH	Pruritus before CLA CLA 22 years earlier than other endocrine symptoms of MEN2A Pruritus relieved temporarily by topical GS
Malhotra et al., 2020 [149]	A 33-year-old	634	1 (100)	0:1 (100)	NA	Right scapular region	NA	Metastatic MTC, PHEO	Pruritus before CLA
Pal R et al., 2018 [150]	A 45-year-old	C634R	1 (100)	0:1 (100)	41	Interscapular region	NA	Metastatic PHEO, MTC	
Zhang XW et al., 2016 [151]	73	634	14/73 ()		NA	NA	NA	NA	
Scapineli JO et al., 2016 [121]	38 ᵃ MEN2A	C634Y/R/W	18 (50)	5:13 (72)	19 ± 10	Interescapular region	NA	MTC 37 (97) PHEO 11 (31) HPTH 6 (19)	−15/18, 83% CLA before other endocrine symptoms of MEN2A Pruritus before CLA
Qi XP et al., 2015 [139]	6 MEN2A	S891A and G513D ᵇ	3 (50)	2:1 (33)	29 ± 2	Upper back and legs, arms, shoulders	All patients were treated with glucocorticoid cream, which resulted in a decreased period of itching, but the application was discontinued due to side effects.	MTC 3 (50) PHEO 0, HPTH 0	Pruritus before CLA
Rothberg AE et al., 2009 [56]	1 FMTC	V804M	1 (100)	0:1 (100)	40	Interscapular	Glucocorticoid cream provided minimal relief, but no other medication was palliative.	MTC 1 (100) PHEO 0, HPTH 0	Pruritus worsened by stress
Gullu S et al., 2005 [144]	1 MEN2A	C634Y	1 (100)	0:1 (100)	34	Interscapular region	NA	1 (100) MTC, 1 (100) bilateral PHEO, 0 (0) HPTH	Pruritus present
Abdullah F et al., 2004 [146]	4 MEN2A	634	3 (75)	0:3 (100)	31 ± 14	Interscapular region	NA	4 (100) MTC, PHEO NA, HPTH NA	
Lemos MC et al., 2003 [152]	5 MEN2A	C634W	4 (80)	1:3 (75)	33 ± 21	NA	NA	5 (100)CMT, 2(40) PHEO, 2(40)HPTH	
Vieira AE et al., 2002 [153]	4 MEN2A	C634Y/R	1 (25)	0:1 (100)	21	NA	NA	1 (100) MTC, 1 (100) PHEO, 1(100) HPTH	
Lemos MC et al., 2002 [154]	1 MEN2A	C634W	1 (100)	NA	5	NA	NA	1 (100) MTC, 0 (0) PHEO, 0 (0) HPTH	
Karga HJ et al., 1998 [145]	29 MEN2A	12 C634R/2 C620Y/15 C634Y	1 (3.4)	1:0 (0)	24	NA	NA	1 (100) MTC, 1 (50) PHEO, 0 (0) HPTH	CLA before the diagnosis of MEN 2A
Seri M et al., 1997 [142]	2 MEN2A	C634G	2 (100)	0:2 (100)	49 ± 7	Interscapular	NA	2 (100) MTC, 1 (50) PHEO, 0 (0) HPTH	Pruritus present
Pacini F et al., 1993 [155]	11 MEN2A	NA	4 (36)	0:4 (100)	NA	Interscapular region	NA	NA	Pruritus present
Kousseff BG et al., 1991 [128]	6 MEN2A	NA	2 (33)	0:2 (100)	NA	Interscapular region	NA	6 (100) 5 (83) 0 (0)	Pruritus present
Ferrer JP et al., 1991 [143]	7 MEN2A	C634W	5 (71)	2:3 (60)	16 ± 7	Interscapular region	NA	7 (100) 0 (0) 0 (0)	
Gagel RF et al., 1989 [136]	5 MEN2A	NA	3 (60)	0:4 (100)	19 ± 1	Upper back	Topical GS	5 (100) 1 (20) NA	Pruritus relieved temporarily by topical GS
Nunziata V et al., 1989 [156]	10 MEN2A	NA	5 (50)	1:4 (80)	7 ± 2	NA	NA	10 (100) 3 (30) NA	Prurit present

**Table 2 ijms-25-09765-t002:** Key findings across our search according to the mentioned references [128,137,139,141,143,146,149,150,151,152,153,154,155,157,158,159,160,161,162].

Number	Key Findings according to Our Search
1.	MEN2-related CLA occurs in patients with C634 *RET* pathogenic variants; most data agree that one-third of the C634-positive subjects have CLA, but the ranges are between 9% and 50%.
2.	One single study showed the *RET* codons map (codon 634 in exon 11) in CLA: C634Y (affecting 43.6% of the subjects), followed by C634R (27.3%), C634W (10.9%), C634G (9.1%), C634F (1.8%), and C634 (5.5%).
3.	Non-C634 germline *RET* pathogenic variants included (at a low level of statistical evidence) the following: *RET* V804M mutation in exon 14 for MTC and CLA (CLA at upper back); *RET* S891A mutation in exon 15 binding *OSMR* variant G513D (FMTC and CLA comprising the lower legs to thighs, upper back, shoulders, arms, and forearms); and C611Y (CLA at interscapular region).
4.	Typically, CLA is detected at an early age (from childhood until young adulthood) before the actual MTC identification unless *RET* screening protocols are already applied.
5.	The time frame between CLA diagnosis and the identification of *RET* pathogenic variants varied between 5 and 60 years according to one study.
6.	Females seem more prone to MEN2-CLA than males.
7.	The same *RET* mutation is not necessarily associated with the same CLA presentation.
8.	In MTC/MEN2 subjects, the most affected CLA area was the scapular region of the upper back.
9.	Alternatively, another hypothesis highlights the fact that CLA is secondary to a long-term prurit/notalgia paresthetica in MTC/MEN2, and not to a distinct *RET* influence.
10.	The relationship between NP and MEN2 might be explained by the fact that neural crest cells, implicated in the embryological development of the adrenal medulla and para-follicular C cells of the thyroid, are also involved in the embryogenesis of the thoracic sensory fibers that explains the pruritus sites.
11.	While the NP hypothesis might explain some cases, pruritus may be largely absent in a high percentage of patients with PLCA, including some patients with LCA-MEN2; thus, other pathogenic loops might actually be involved.
12.	Alternatively, it has been suggested that pathogenic variants in *OSMR* and *IL31RA* genes lead to incorrect IL-31 signaling, which is directly related to local pruritus (and the co-presence of other non-RET mutations should be taken into consideration to explain CLA in MTC/MEN2).
13.	*OSMR* p.G513D may play a role in modifying the evolutionary processes of LA in the subjects that co-harbor *RET* mutations (further studies are necessary to sustain this aspect).
14	Awareness in CLA-positive patients is essential, including the decision of *RET* testing in selected cases.

## Data Availability

Not applicable.

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
