# Peer review of "Thyroid Malignancy and Cutaneous Lichen Amyloidosis: Key Points Amid RET Pathogenic Variants in Medullary Thyroid Cancer/Multiple Endocrine Neoplasia Type 2 (MEN2)"

_ijms, 2024, doi:10.3390/ijms25189765_

Round 1

Reviewer 1 Report

Comments and Suggestions for Authors

Review to manuscript ijms-3168965

The manuscript provides a comprehensive historical, clinical and molecular landscape on the RET pathogenic variants and their implications in the diagnosis of medullary thyroid cancer/multiple endocrine neoplasia type 2, particularly with respect to the presence of the cutaneous lichen amyloidosis.

The review is long and complex and I would advise the authors to try to streamline it if they feel it is possible without diminishing the information it contains.

Below are my comments and changes to the text.

·         The use of the standard nomenclature according to the recommendations of the Human Genome Variation Society (HGVS) would be advisable for proper reporting of variants, but in the context of a review, commonly used colloquial nomenclature may also be accepted in order not to burden the text.

·         “Amid”, “kindred”, “ailments”: Are these terms in common use? Please consider using a synonym.

·         Line 129: “so-called” sporadic MTC please add (sMTC).

·         Line 312-313 and 342-343: Cite ref 141 only once.

·         Line 314-315. Cite ref 52 only once.

·         Line 316-318. Cite ref 142 only once.

·         Line 324-326. Cite ref 56 only once.

·         Line 327-328. Cite ref 143 only once.

·         Line 328-331 and 346-347: Cite ref 144 only once.

·         The format of Table 1 is difficult to read, please provide a better format.

·         Line 404-406. Cite ref 162 only once.

·         Line 448-450. Cite ref 180 only once.

·         Line 521: the word respectively should be positioned in a way it can be understood

·         In the conclusion the list of key points should be the shorter as possible in order to give a concise and clear take-home message.

Comments on the Quality of English Language

The English seems to me understandable and the text, despite its length, is comprehensible and clear. I only noticed some unusual terms which I have reported in the comments.

Author Response

Response to Review 1 Comments

Dear Reviewer,

Thank you very much for your time and your effort to review our manuscript.

We are very grateful for providing your valuable feedback on the article.

Here is our response and related amendment that has been made in the manuscript according to your review (marked in green color).

The manuscript provides a comprehensive historical, clinical and molecular landscape on the RET pathogenic variants and their implications in the diagnosis of medullary thyroid cancer/multiple endocrine neoplasia type 2, particularly with respect to the presence of the cutaneous lichen amyloidosis.

Thank you very much. We really appreciate it.

The review is long and complex and I would advise the authors to try to streamline it if they feel it is possible without diminishing the information it contains.

Thank you very much. We revisited the manuscript. Thank you

Below are my comments and changes to the text.

Thank you very much. We addressed them as follows:

The use of the standard nomenclature according to the recommendations of the Human Genome Variation Society (HGVS) would be advisable for proper reporting of variants, but in the context of a review, commonly used colloquial nomenclature may also be accepted in order not to burden the text.

Thank you very much. Indeed, we confirm, our approach has a multidisciplinary practical perspective. Thank you

“Amid”, “kindred”, “ailments”: Are these terms in common use? Please consider using a synonym.

Thank you very much. We replaced them. Thank you

Line 129: “so-called” sporadic MTC please add (sMTC).

Thank you very much. We corrected it.

Line 312-313 and 342-343: Cite ref 141 only once.

Thank you very much. We corrected it.

Line 314-315. Cite ref 52 only once.

Thank you very much. We corrected it.

Line 316-318. Cite ref 142 only once.

Thank you very much. We corrected it.

Line 324-326. Cite ref 56 only once.

Thank you very much. We corrected it.

Line 327-328. Cite ref 143 only once.

Thank you very much. We corrected it.

Line 328-331 and 346-347: Cite ref 144 only once.

Thank you very much. We corrected it.

The format of Table 1 is difficult to read, please provide a better format.

Thank you very much. We corrected it.

Line 404-406. Cite ref 162 only once.

Thank you very much. We corrected it.

Line 448-450. Cite ref 180 only once.

Thank you very much. We corrected it.

Line 521: the word respectively should be positioned in a way it can be understood

Thank you very much. We corrected it.

In the conclusion the list of key points should be the shorter as possible in order to give a concise and clear take-home message.

Thank you very much. We reduced them and introduced Table 2 to make it clear according to your recommendations. Thank you

Comments on the Quality of English Language. The English seems to me understandable and the text, despite its length, is comprehensible and clear. I only noticed some unusual terms which I have reported in the comments.

Thank you very much. We replaced them. Thank you

Thank you very much.

Reviewer 2 Report

Comments and Suggestions for Authors

IJMS-3168965:

Thyroid malignancy and cutaneous lichen amyloidosis: key points amid RET pathogenic variants in medullary thyroid cancer/multiple endocrine neoplasia type 2 (MEN2), authored by Stanescu, et al.

General comments:

The authors aimed to update narrative review with respect to the RET pathogenic variants and their implications at clinical and molecular level amid the diagnosis of MTC/MEN2 by focusing on the cutaneous lichen amyloidosis (CLA) using PubMed search.  The authors concluded that CLA in thyroid cancer, particularly, MTC had several key characteristics including the RET gene mutations, implying the importance of awareness in CLA-positive patients and testing for RET gene mutation in the selected cases.  This review manuscript includes rare but important clinical issues; however, the structures and conclusions should be improved for the readers to clarify the points. 

Specific comments:

1. Abstract includes so many points regarding the interrelationships between the RET variants, MTC/MEN2 and the current focus of CLA.  The evidence should be more concise and clearer, and the same points should be amended in the conclusion as well.

2. The structure of the whole manuscript should be modified to clarify the points in each section.  It should be started from the clinical significance of the RET gene, and then CLA issues, and the paragraphs on the MTC/MEN2 would be following the RET/CLA issues.

3. The main results of Table 1 should be rearranged to be understood easily by the various readers.  The reported country and periods, the examined modality, hormonal data and followed durations should also be added.  Some parentheses are empty and “?” should be “not available”.

4. A couple of figures showing the interrelationships among the RET variants, CLA and MTC/MEN2 should be added in the revised manuscript to explain the connection of each pathology and clinical significance.  Since the manuscript is quite descriptive, the edges of the previous knowledge and the updated points should be marked in the figure.

5. Also, the genetic and pathological assessments seem not sufficient in this manuscript.  The reported locations of RET variants related to CLA pathology with or without MTC/MEN2 should be summarized.  In the pathological aspect, the representative figures of skin biopsy of the CLA specimen would be shown, if possible, in the cases that have RET mutations and/or MTC/MEN2 complications.

6. The number of references is too many for the length of the text of this manuscript.  The selection considering the level of evidence should be carefully done for this review paper.

Author Response

Response to Review 2 Comments

Dear Reviewer,

Thank you very much for your time and your effort to review our manuscript.

We are very grateful for your insightful comments and observations, also, for providing your valuable feedback on the article.

Here is a point-by-point response and related amendments that have been made in the manuscript according to your review (marked in green color).

Thyroid malignancy and cutaneous lichen amyloidosis: key points amid RET pathogenic variants in medullary thyroid cancer/multiple endocrine neoplasia type 2 (MEN2), authored by Stanescu, et al.

General comments:

The authors aimed to update narrative review with respect to the RET pathogenic variants and their implications at clinical and molecular level amid the diagnosis of MTC/MEN2 by focusing on the cutaneous lichen amyloidosis (CLA) using PubMed search.  The authors concluded that CLA in thyroid cancer, particularly, MTC had several key characteristics including the RET gene mutations, implying the importance of awareness in CLA-positive patients and testing for RET gene mutation in the selected cases.  This review manuscript includes rare but important clinical issues; however, the structures and conclusions should be improved for the readers to clarify the points. 

Thank you very much. We really appreciate it.

Specific comments:

  1. Abstract includes so many points regarding the interrelationships between the RET variants, MTC/MEN2 and the current focus of CLA. The evidence should be more concise and clearer, and the same points should be amended in the conclusion as well.

Thank you very much. We respectfully mention that indeed the topic is complex and the panel of approach is large. We followed your recommendations and reduced the abstract and the conclusion. As main key points-related conclusion we introduced a novel Table (Table 2). Thank you

  1. The structure of the whole manuscript should be modified to clarify the points in each section.  It should be started from the clinical significance of the RET gene, and then CLA issues, and the paragraphs on the MTC/MEN2 would be following the RET/CLA issues.

Thank you very much. We respectfully mention that MTC/MEN2 is the clinical expression of the RET anomalies and this clinical picture might associate the CLA issue amid the entire MTC/MEN2 panel, that it why we opted for this structure across a narrative review which allows a more flexible approach. Thank you

  1. The main results of Table 1 should be rearranged to be understood easily by the various readers.  The reported country and periods, the examined modality, hormonal data and followed durations should also be added.  Some parentheses are empty and “?” should be “not available”.

Thank you very much. We corrected the table. The hormonal panel is reflected by the endocrine diagnosis which is included in a distinct column. The specific hormonal assays (which varied over time depending on the clinical presentation and associated management) and durations are out of our scope and the data are already complex and multiple. We corrected “NA”. Thank you.

  1. A couple of figures showing the interrelationships among the RET variants, CLA and MTC/MEN2 should be added in the revised manuscript to explain the connection of each pathology and clinical significance.  Since the manuscript is quite descriptive, the edges of the previous knowledge and the updated points should be marked in the figure.

Thank you very much. We followed your recommendations and introduced novel figures. Thank you

  1. Also, the genetic and pathological assessments seem not sufficient in this manuscript.

 The reported locations of RET variants related to CLA pathology with or without MTC/MEN2 should be summarized. In the pathological aspect, the representative figures of skin biopsy of the CLA specimen would be shown, if possible, in the cases that have RET mutations and/or MTC/MEN2 complications.

Thank you very much. We provided all the genetic and pathological data we were able to find as mentioned by the methods. Other areas are yet to be identified as mentioned, too. We summarized the data across a novel table. This is a narrative review, we respectfully mention that no skin biopsy/case report is at our hand at this point, neither it represents our purpose in this article which is a literature review, and not a case study/series. Thank you

  1. The number of references is too many for the length of the text of this manuscript.  The selection considering the level of evidence should be carefully done for this review paper.

Thank you very much. We reduced them. We respectfully mention that MDPI rules do not limit the length, neither the number of references of an article, and, as mentioned, the level of evidence is not abundant in many areas as far as we know at this point. Thank you

Thank you very much.

Round 2

Reviewer 2 Report

Comments and Suggestions for Authors

The authors revised and improved their manuscript appropriately.  However, we recommend to reduce the results (still 35%) of crosscheck by iThenticate.

Author Response

Response to Review 2 Comments

Dear Reviewer,

Thank you very much for your time and your effort to review our manuscript.

We are very grateful for providing your valuable feedback on the article.

Here is our response and related amendment that has been made in the manuscript according to your review.

The authors revised and improved their manuscript appropriately.  However, we recommend to reduce the results (still 35%) of crosscheck by iThenticate.

Thank you very much. We adjusted it according to your recommendation.

Thank you very much